# Privacy-Preserving Approach for Early Detection of Long-Lie Incidents: A Pilot Study with Healthy Subjects

**DOI:** 10.3390/s25123836

**Published:** 2025-06-19

**Authors:** Riska Analia, Anne Forster, Sheng-Quan Xie, Zhiqiang Zhang

**Affiliations:** 1School of Electronic and Electrical Engineering, University of Leeds, Leeds LS2 9JT, UK; elran@leeds.ac.uk (R.A.); s.q.xie@leeds.ac.uk (S.-Q.X.); 2Department of Electrical Engineering, Politeknik Negeri Batam, Batam 29461, Indonesia; 3Academic Unit for Ageing and Stroke Research, Leeds Institute of Health Sciences, University of Leeds, Leeds LS2 9LN, UK; a.forster@leeds.ac.uk

**Keywords:** long-lie detection, thermal imaging, ensemble learning, privacy-preserving monitoring, edge computing

## Abstract

(1) Background: Detecting long-lie incidents—where individuals remain immobile after a fall—is essential for timely intervention and preventing severe health consequences. However, most existing systems focus only on fall detection, neglect post-fall monitoring, and raise privacy concerns, especially in real-time, non-invasive applications; (2) Methods: This study proposes a lightweight, privacy-preserving, long-lie detection system utilizing thermal imaging and a soft-voting ensemble classifier. A low-resolution thermal camera captured simulated falls and activities of daily living (ADL) performed by ten healthy participants. Human pose keypoints were extracted using MediaPipe, followed by the computation of five handcrafted postural features. The top three classifiers—automatically selected based on cross-validation performance—formed the soft-voting ensemble. Long-lie conditions were identified through post-fall immobility monitoring over a defined period, using rule-based logic on posture stability and duration; (3) Results: The ensemble model achieved high classification performance with accuracy, precision, recall, and an F1 score of 0.98. Real-time deployment on a Raspberry Pi 5 demonstrated the system is capable of accurately detecting long-lie incidents based on continuous monitoring over 15 min, with minimal posture variation; (4) Conclusion: The proposed system introduces a novel approach to long-lie detection by integrating privacy-aware sensing, interpretable posture-based features, and efficient edge computing. It demonstrates strong potential for deployment in homecare settings. Future work includes validation with older adults and integration of vital sign monitoring for comprehensive assessment.

## 1. Introduction

Long-lie incidents—conditions in which individuals, especially older adults, remain immobile on the floor for extended periods after a fall—pose serious health risks. These include dehydration, pressure ulcers, infections, and even death in severe cases [1,2,3]. Studies show that nearly 98% of long-lie events occur when individuals are alone at home, leading to delayed medical assistance and increased morbidity [4]. Timely and accurate identification of such events is therefore critical to improve patient outcomes and reduce healthcare burdens [2,5].

While fall detection systems have been widely investigated [6,7,8,9,10], the specific identification of long-lie conditions remains limited. Unlike fall detection, which focuses on the immediate impact event, long-lie detection requires sustained post-fall monitoring to assess whether a person remains incapacitated or unable to seek help. Clinical studies have suggested thresholds ranging from 24.5 s to over an hour as critical indicators for long-lie scenarios, highlighting their association with increased morbidity and mortality [2,5,11,12,13]. Moreover, some research indicates that long-lie events may occur independently of falls or emerge gradually, further complicating early recognition [3]. The absence of a standardized definition and the diversity of post-fall behaviors—such as changes in posture, partial movements, and occlusions—present additional challenges to automatic detection [3,14].

To address these complexities, various sensing modalities have been proposed. Wearable-based systems using accelerometers or gyroscopes have shown promise in controlled environments [15,16], but their practical use is often hindered by user discomfort, low compliance, and stigma, particularly among older adults [4,17]. Meanwhile, vision-based methods using RGB cameras provide rich contextual information and have demonstrated reductions in Time On Ground (TOG) in controlled trials [18]. However, such approaches raise privacy concerns and suffer from degraded performance under poor lighting conditions [19,20]. Thermal imaging has emerged as a privacy-preserving and lighting-agnostic alternative for human activity monitoring [21,22,23]. By capturing heat signatures rather than identifiable visuals, thermal systems mitigate privacy issues while maintaining robustness in real-world conditions. When integrated with edge computing platforms such as the Raspberry Pi, they offer the potential for low latency and real-time inference without relying on external servers [24,25]. Despite these promising capabilities, very few existing systems are designed specifically for long-lie detection, and even fewer are optimized for lightweight, on-device computation using privacy-sensitive inputs. Most previous work remains limited to detecting fall events alone, overlooking the critical need to assess post-fall recovery attempts or prolonged immobility. Additionally, current machine learning approaches often rely on resource-intensive architectures that are not feasible for embedded deployment or lack clinical insight into how long-lie conditions manifest in practice [7,26,27,28].

To address these limitations, we propose a novel, real-time, and privacy-preserving long-lie detection system that leverages thermal imaging and a lightweight soft-voting ensemble classifier. The system introduces a dual-stage detection mechanism that not only identifies fall events but also monitors post-fall immobility using interpretable, handcrafted posture features. By utilizing thermal imaging and geometric-based feature extraction, the approach ensures privacy preservation without relying on identifiable visuals or complex deep learning architectures. Moreover, the system is designed for efficient deployment on low-power edge devices such as the Raspberry Pi 5, enabling real-time monitoring without dependence on cloud infrastructure. The model is validated using a custom-built dataset collected specifically for this study from ten healthy participants performing both activities of daily living (ADL) and simulated fall scenarios. This self-collected dataset, recorded exclusively through thermal imaging under controlled indoor conditions, ensures data quality, privacy, and relevance to the problem domain. Altogether, this study presents a practical and interpretable solution for the early detection of long-lie incidents, contributing to the development of intelligent, privacy-aware assistive technologies that support safe and independent living for older adults.

The remainder of this paper is structured as follows: previous research that relates to this work is explained in Section 2. Section 3 details the methodology, including data collection, feature extraction, and model design. Section 4 presents experimental results. Section 5 discusses the system’s performance and deployment feasibility. Finally, Section 6 concludes with insights and future directions, including the integration of vital sign monitoring.

## 2. Related Work

Thermal imaging-based fall detection has emerged as a promising, privacy-preserving alternative to RGB and depth-based methods, particularly for in-home monitoring of older adults. Prior studies have shown that thermal vision can effectively capture postural and motion anomalies [21,22,29]. While thermal cameras protect user anonymity and function in low-light conditions, their limited resolution and lack of texture detail present significant challenges for accurate pose estimation. Sensor-related issues—such as calibration drift and thermal diffusion—can further degrade keypoint localization, especially under occlusion or changing ambient temperatures. These limitations are particularly problematic when estimating posture-dependent features, such as body orientation or width-to-height ratio (WHR), which are critical for identifying abnormal immobility.

Recent work has explored pose-based fall classification using thermal imagery with enhanced architectures and descriptors to improve detection under low-light or infrared conditions [30,31]. These systems typically extract postural dynamics—such as bounding box elongation, centroid motion, or keypoint confidence—to distinguish falls from regular activity. Similarly, efforts to develop lightweight pose estimation models for thermal images, such as THPoseLite, have significantly improved keypoint detection under limited computational resources [32]. However, despite advancements in classification accuracy and model efficiency, these approaches are limited to instantaneous fall detection. None incorporate temporal reasoning to monitor post-fall states or identify prolonged immobility. As such, the critical problem of long-lie detection—an essential indicator of post-fall morbidity—remains unaddressed mainly in thermal-based monitoring systems.

Deep learning models, including 3D CNNs and recurrent architectures, have demonstrated strong performance in thermal-based fall detection [20,29,33]. However, these models typically require substantial computational resources, making them unsuitable for deployment on low-power embedded systems. Hybrid architectures combining autoencoders with deep convolutional models have also been proposed; however, their complexity and reliance on paired modalities limit their practical deployment [29,34]. To address deployment constraints, classical machine learning approaches using handcrafted features have been explored on low-resolution thermal data [16,35]. Although these methods are more lightweight, they often suffer from reduced robustness under varying ambient conditions. Furthermore, while some systems demonstrate feasibility in real-world use, many still lack interpretability and scalability—essential factors for long-term deployment in resource-limited environments [20,36]. Others have achieved high accuracy using coarse-resolution infrared arrays [33] but fall short of addressing post-fall monitoring or immobility assessment.

Critically, the majority of existing thermal-based systems focus solely on detecting the initial fall event [21,26] without addressing the equally important problem of long-lie detection—prolonged immobility after a fall, which is a key risk factor for morbidity and mortality in elderly care. Although recent approaches in neuromorphic and event-based vision offer temporal fall localization capabilities [19], they often rely on specialized sensors that are not suited for widespread deployment. In addition to technical gaps, ethical considerations remain a central concern. Continuous monitoring systems must be designed with transparency and user consent in mind, especially when deployed in private environments such as bedrooms or care homes [37].

This work presents the first thermal-based fall detection framework that explicitly addresses long-lie detection as a primary objective. Unlike prior systems that focus solely on identifying fall events, the proposed method introduces a rule-based temporal logic module to monitor post-fall immobility—an essential indicator of health risk in elderly care. The system operates entirely on thermal input, preserving privacy, and employs interpretable handcrafted features with a soft-voting ensemble classifier, which is designed for real-time execution on edge devices, offering a lightweight and deployable solution tailored for continuous in-home monitoring. By integrating long-lie detection within a privacy-preserving architecture, this approach fills a critical gap in existing literature.

## 3. Materials and Methods

Some studies have found that long-lie incidents are often caused by falls, leaving individuals immobilized on the ground for extended durations exceeding one hour [2,11,12,13]. Motivated by these findings, this study presents a system that detects long-lie events by analyzing characteristic patterns of post-fall immobility. We propose a real-time, privacy-preserving detection framework that utilizes thermal imaging in combination with a soft-voting ensemble classifier. The system is specifically designed for efficient operation on edge devices, achieving high detection accuracy while addressing privacy concerns.

The pipeline process, illustrated in Figure 1, includes four primary stages: (1) thermal data acquisition and pose estimation, (2) preprocessing and temporal data augmentation, (3) handcrafted feature extraction, and (4) classification using a lightweight ensemble model. The system also incorporates a long-lie detection logic that monitors postural stability over time following a fall event. A custom dataset was collected specifically for this study, consisting of simulated fall and ADL sequences recorded solely using thermal imaging to ensure privacy and realism.

### 3.1. Thermal Data Collection

To support privacy-aware and real-time long-lie detection on edge devices, this study employs a low-resolution (256 × 192) thermal camera mounted on an adjustable tripod. The camera can be extended up to 170 cm in height and is angled downward at approximately 30 degrees, providing an elevated frontal view. This setup ensures unobstructed capture of full-body movements during falls and daily activities while maintaining portability and ease of deployment in indoor environments.

Data collection was conducted in a controlled indoor setting using thermal imaging exclusively to preserve privacy. Ten healthy adult participants (five male, five female), aged 29.4±4.52 years, with an average height of 164.4±9.34 cm and weight of 63.4±7.90 kg, were recruited for the study. Each participant performed 32 types of simulated falls and 15 ADLs, resulting in a total of 320 fall and 150 ADL video sequences. Participant demographics are summarized in Table 1.

### 3.2. Pose-Based Feature Extraction and Dataset Preparation

The dataset development pipeline is illustrated in Figure 2. It begins with thermal video capture, followed by keypoint extraction using an efficient implementation of the MediaPipe Pose estimation algorithm v0.10.9 (Google LLC, Mountain View, CA, USA), which is compatible with CPU-based edge devices such as the Raspberry Pi 5 (Raspberry Pi Foundation, Cambridge, UK) [38]. Given the inherent limitations of thermal imaging—such as reduced contrast and occasional occlusions—a lightweight adaptation of the original pose estimation framework [39] was applied, specifically optimized for thermal modalities. To enhance reliability, the extraction focused on a subset of 13 keypoints that consistently exhibited clear visibility across sequences and contributed meaningfully to postural representation.

This subset includes upper body landmarks (nose, shoulders, elbows, and wrists), as well as hips, knees, and ankles from the lower body. While lower-limb keypoints are sometimes occluded in side-view configurations, their inclusion—particularly hips and knees—has been found to improve the computation of posture-related features, such as orientation angles and the width-to-height ratio (WHR). Confidence-based filtering and selective use of keypoints helped reduce noise from unreliable detections. Temporal smoothing through backward interpolation was further applied to mitigate transient dropout and ensure sequence continuity.

The extracted keypoints were then preprocessed, augmented to increase data diversity, and used for handcrafted feature extraction. The final feature sets were saved in CSV format for model training and evaluation.

#### 3.2.1. Data Preprocessing

The data preprocessing stage comprises two main components: data cleaning and sample selection, as outlined in the dataset generation pipeline. This step is essential for ensuring pose data quality and enabling reliable long-lie detection, particularly in the context of thermal imaging and low-power edge deployment limitations.

To mitigate pose estimation noise on thermal frames, a confidence-based filtering strategy was applied. Specifically, keypoints with detection confidence below 0.5 were excluded from further processing. A temporal interpolation mechanism was also employed to restore short-term missing keypoints and maintain spatial continuity. Although this study did not include manual benchmarking against annotated thermal pose datasets, we performed a detection consistency analysis to assess the reliability of keypoint detection. We evaluated a representative subset of seven keypoints—nose, shoulders, hips, and knees—using two thermal video recordings captured at a resolution of 256 × 192.

The detection rate for each keypoint was computed as the percentage of frames in which the keypoint was successfully detected (i.e., confidence > 0.5) relative to the total number of frames:(1)$$\mathrm{Detection}\;\mathrm{Rate}\;(\%)=\left(\frac{\mathrm{valid}\;\mathrm{frames}}{\mathrm{total}\;\mathrm{frames}}\right)\times 100$$

Table 2 summarizes the aggregated detection results. While upper-body and torso keypoints achieved consistently high detection rates above 99%, knees exhibited more variability, likely due to occlusion or pose-related thermal distortion. The overall average detection rate across keypoints was 93.96%, with a standard deviation of 8.22, confirming the robustness of the selected keypoint subset under thermal conditions.

*(1) Data Cleaning:* This process enhances the reliability and temporal consistency of skeleton keypoints extracted from thermal video frames. Compared with RGB-based inputs, thermal data often produces noisier or incomplete keypoints due to lower spatial contrast and sensor artifacts. To address these challenges, a two-step cleaning strategy is employed: (i) removal of frames with insufficient or missing keypoints and  (ii) replacement of missing keypoints using temporally adjacent values.

Let the keypoint dataset be defined as(2)$$D=\{k_{i}|i\in I\}$$
where $k_{i}=\left(k_{i1},k_{i2},\ldots ,k_{im}\right)$ represents the *m* keypoints in frame *i*, and I denotes the set of all time indices.

**Step 1: Elimination of invalid keypoints.** A frame is discarded if any required keypoint $k_{ij}$ remains undetected for longer than the predefined threshold $T_{1}$:(3)$$D^{*}=\left\{k_{i}\in D|k_{ij}\neq ⌀\;\mathrm{within}\;T_{1}\right\}$$
where *⌀* indicates a missing keypoint value.

**Step 2: Temporal replacement of missing values.** Transient missing values—often caused by motion blur or momentary occlusion—are replaced using the corresponding keypoints from the previous valid frame within the threshold $T_{2}$:(4)$$k_{i}=\left\{\begin{array}{cc} k_{i-1} & \mathrm{if}\;k_{i}\;\mathrm{has}\;\mathrm{missing}\;\mathrm{keypoints}\;\mathrm{within}\;T_{2}\\ k_{i} & \mathrm{otherwise} \end{array}\right.$$

This approach was tailored specifically for thermal skeleton data, enabling robust temporal correction without re-estimation. The method is computationally efficient and well-suited for real-time applications on resource-constrained devices.

*(2) Sample Selection:* Following cleaning, the skeleton sequences are segmented using a sliding window approach with fixed-length overlapping segments. Each segment comprises 40 frames (equivalent to 4 s at 100 ms per frame). This window length was selected based on clinical findings showing that most fall events in older adults—from loss of balance to impact—occur within a 4-s period [40]. Capturing this time interval is crucial for distinguishing falls from other daily activity transitions.

From each segment, five handcrafted features are computed: the width-to-height ratio, three body orientation angles ($\theta_{1},\theta_{2},\theta_{3}$), and the angular center of body orientation (θ). The temporal sample is structured as(5)$$\left\{x_{i}\right\}=\left(x_{1},x_{2},x_{3},\ldots ,x_{40}\right)$$(6)$$\begin{array}{c} \left[\begin{array}{cccc} x_{1}^{1} & x_{2}^{1} & \ldots & x_{40}^{1}\\ x_{1}^{2} & x_{2}^{2} & \ldots & x_{40}^{2}\\ x_{1}^{3} & x_{2}^{3} & \ldots & x_{40}^{3}\\ x_{1}^{4} & x_{2}^{4} & \ldots & x_{40}^{4}\\ x_{1}^{5} & x_{2}^{5} & \ldots & x_{40}^{5} \end{array}\right]=\left[\begin{array}{c} X_{1}\\ X_{2}\\ X_{3}\\ X_{4}\\ X_{5} \end{array}\right]=X\in R^{5\times 40} \end{array}$$(7)$$V=(X_{1},X_{2},X_{3},\ldots ,X_{T})$$
where $X_{T}$ is the temporal feature matrix of segment *T*, and V is the sequence of such segments. While the structure of window-based segmentation follows previous practices in pose-based activity recognition [41], the current work extends it for thermal keypoint data and adapts it with a lightweight, handcrafted feature set optimized for thermal-based fall detection and privacy-aware embedded systems.

#### 3.2.2. Temporal Data Augmentation

To address the class imbalance and enhance the temporal diversity of motion patterns, a tailored temporal data augmentation strategy was implemented based on overlapping sliding window segmentation. Let *W* represent a video sequence segmented into overlapping clips $D_{i}$ of fixed length *M*, where the degree of overlap is defined by the ratio:(8)$$\beta =\frac{M_{\mathrm{overlap}}}{M}$$

To account for differing motion dynamics between fall and ADL events, distinct overlap rates were applied: $\beta_{1}$ for fall sequences and $\beta_{2}$ for ADL sequences. The number of augmented clips for each category is given by(9)$$N_{\mathrm{clips}\_\mathrm{fall}}=N_{f}\left(1+\frac{1}{\beta_{1}}\right),\;N_{\mathrm{clips}\_\mathrm{ADL}}=N_{\mathrm{adl}}\left(1+\frac{1}{\beta_{2}}\right)$$

The total number of augmented samples across all participants *P* is expressed as(10)$$N_{\mathrm{total}}=P\cdot \left(N_{\mathrm{clips}\_\mathrm{fall}}+N_{\mathrm{clips}\_\mathrm{ADL}}\right)$$

This adaptive augmentation approach is designed specifically for fall detection under thermal imaging, where class imbalance and motion ambiguity are common. Unlike conventional augmentation techniques that apply uniform strategies, our method introduces differentiated overlap parameters to better preserve contextual motion cues for both fall and non-fall activities. As a result, a total of 2756 temporally enriched and labeled sequences were generated, substantially improving the balance and representativeness of the dataset while mitigating the risks of overfitting.

#### 3.2.3. Handcrafted Feature Extraction

Following temporal segmentation and data augmentation, five handcrafted geometric features were extracted from each sample segment to capture posture-related variations between fall events and daily activities. These features were specifically designed to address the limitations of thermal keypoints, which often suffer from reduced spatial resolution and partial occlusions. The feature set includes the width-to-height ratio, body orientation angles, and the angular center of body posture—all derived from thermal pose estimation [27,28]. These compact and interpretable features were chosen for their effectiveness in characterizing both the dynamics of falling and the patterns of sustained immobility, making them well-suited for real-time detection in privacy-sensitive applications.

*(1) Width-to-Height Ratio (WHR):* To describe overall body compactness, a 2D posture-based width-to-height ratio was computed, as illustrated in Figure 3a. Unlike prior work [42,43] that used full-body joint sets, our formulation used only the shoulder and knee keypoints to enhance reliability in thermal images. The ratio is calculated as(11)$$W=\max\left(x_{s},x_{k}\right)-\min\left(x_{s},x_{k}\right),\;H=\max\left(y_{s},y_{k}\right)-\min\left(y_{s},y_{k}\right)$$(12)$$\mathrm{WHR}=\frac{W}{H}$$
where $\left(x_{s},y_{s}\right)$ and $\left(x_{k},y_{k}\right)$ denote the shoulder and knee coordinates, respectively.

*(2) Body Angle Orientation:* This feature evaluates postural orientation, distinguishing between upright and supine positions by calculating the center of gravity (COG) for the upper, whole, and lower body segments, as shown in Figure 3b. The COG for a body segment (BS) is calculated as(13)$${\mathrm{COG}}_{\mathrm{BS}}\left[x\right]=\frac{\sum_{i=1}^{n}m_{i}x_{i}}{\sum_{i=1}^{n}m_{i}},\;{\mathrm{COG}}_{\mathrm{BS}}\left[y\right]=\frac{\sum_{i=1}^{n}m_{i}y_{i}}{\sum_{i=1}^{n}m_{i}}$$
where $m_{i}$ is the mass, which refers to the area of image height and width that is covered on the segment area, and $x_{i}$, $y_{i}$ are the coordinates of the *i*-th segment. To calculate the COG based on body location, what is needed is(14)$${\mathrm{COG}}_{\mathrm{body}}\left[x,y\right]=\frac{\sum {\mathrm{COG}}_{\mathrm{BS}}\left[x,y\right]}{\sum \mathrm{body}\;\mathrm{segment}}$$
where $\sum {\mathrm{COG}}_{\mathrm{BS}}\left[x,y\right]$ is the sum of the COGs for all body segments.

To determine the body angle orientation, we used the COGs for the upper body ($C_{\mathrm{UB}}$), whole body ($C_{\mathrm{WB}}$), and lower body ($C_{\mathrm{LB}}$). As a person falls, these COGs shift from vertical to horizontal alignment, as shown in Figure 3c, indicating different postural states (standing, nearly fallen, or fallen). The angles between COGs were calculated as follows:(15)$$\begin{array}{cc} \theta_{1} & =\arctan\left(\left(C_{{\mathrm{UB}}_{y}}-C_{{\mathrm{WB}}_{y}}\right),\left(C_{{\mathrm{UB}}_{x}}-C_{{\mathrm{WB}}_{x}}\right)\right)\\ \; & \;-\arctan\left(\left(y_{y}^{\prime }-C_{{\mathrm{WB}}_{y}}\right),\left(C_{{\mathrm{WB}}_{x}}-C_{{\mathrm{WB}}_{x}}\right)\right)\\ \theta_{2} & =\arctan\left(\left(C_{{\mathrm{UB}}_{y}}-C_{{\mathrm{LB}}_{y}}\right),\left(C_{{\mathrm{UB}}_{x}}-C_{{\mathrm{LB}}_{x}}\right)\right)\\ \; & \;-\arctan\left(\left(y_{y}^{\prime }-C_{{\mathrm{LB}}_{y}}\right),\left(C_{{\mathrm{LB}}_{x}}-C_{{\mathrm{LB}}_{x}}\right)\right)\\ \theta_{3} & =\arctan\left(\left(C_{{\mathrm{WB}}_{y}}-C_{{\mathrm{LB}}_{y}}\right),\left(C_{{\mathrm{WB}}_{x}}-C_{{\mathrm{LB}}_{x}}\right)\right)\\ \; & \;-\arctan\left(\left(y_{y}^{\prime }-C_{{\mathrm{LB}}_{y}}\right),\left(C_{{\mathrm{LB}}_{x}}-C_{{\mathrm{LB}}_{x}}\right)\right) \end{array}$$

*(3) Angle Center of Body Orientation (ACB (θ)):* The ACB angle measures the vertical and horizontal angles of the COGs during movement, helping estimate the angular displacement before falling. Figure 3d shows the ACB angle with the lower body COG as a blue dot at $c_{\left(x,y\right)}$, the hip center as a green dot at $b_{\left(x,y\right)}$, and the upper body COG as an orange dot at $a_{\left(x,y\right)}$. Angular deviation occurs during a fall because of the displacement of these points. The hip center coordinates $C_{\mathrm{hip}}\left[x,y\right]$ are calculated as follows:(16)$$\begin{array}{cc} x_{c\_\mathrm{hip}} & =\frac{x_{\mathrm{lh}}+x_{\mathrm{rh}}}{2}\\ y_{c\_\mathrm{hip}} & =\frac{y_{\mathrm{lh}}+y_{\mathrm{rh}}}{2}\\ C_{\mathrm{hip}}\left[x,y\right] & =(x_{c\_\mathrm{hip}},\;y_{c\_\mathrm{hip}}) \end{array}$$
while the COG of the lower body and upper body was obtained using Equation (Equation 14). After determining the COGs of the upper body, knee, and hip center coordinates, the ACB angle (θ) was calculated as follows:(17)$$\begin{array}{cc} \theta_{\mathrm{rad}} & =\arctan\left(\left(c_{y}-b_{y}\right),\left(c_{x}-b_{x}\right)\right)-\arctan\left(\left(a_{y}-b_{y}\right),\left(a_{x}-b_{x}\right)\right)\\ \theta_{\deg} & =\mathrm{abs}\left(\frac{\theta_{\mathrm{rad}}\times 180}{\pi }\right) \end{array}$$

This feature set was engineered explicitly for thermal-based pose estimation, where conventional skeleton representations may be incomplete or noisy. By prioritizing angular and proportional cues from reliable keypoints, the extracted features offer enhanced discriminative power for lightweight and privacy-aware fall detection systems. Moreover, the compact nature of these features makes them highly suitable for real-time inference on resource-constrained edge devices such as Raspberry Pi, ensuring fast and energy-efficient deployment in practical homecare scenarios.

To address the impact of camera viewpoint on keypoint recognition and body parameter estimation, all data in this study were recorded using a fixed 30° side-view angle with a consistent subject-to-camera distance of approximately 1.8 to 2.0 m. This setup minimizes perspective distortion and ensures stable keypoint projection geometry across samples. Additionally, z-score normalization was applied to all features to reduce inter-subject variability in scale. To evaluate the robustness of the WHR, body angle orientation, and ACB under this setup, we analyzed 2755 samples across all postural features. The mean projected WHR was 1.94 ± 2.53, while body angle orientations $\theta_{1}$, $\theta_{2}$, and $\theta_{3}$ each showed a consistent mean of 0.72 radians with standard deviations around 0.69–0.70. The ACB angle (θ) exhibited a mean of 3.02 radians with a standard deviation of 0.66, consistent with horizontal orientation under long-lie or supine conditions. These results indicate that the observed feature variations primarily reflect authentic postural transitions rather than camera-induced artifacts. This supports the use of a fixed-angle 2D setup for reliable WHR and orientation estimation in controlled environments. However, we acknowledge that future real-world deployments may require depth-based compensation or view-invariant feature enhancement to maintain accuracy across varied camera angles.

#### 3.2.4. Dataset Generation

All extracted features from each sample segment were compiled into a structured CSV format to serve as inputs for the soft-voting ensemble classifier. Each segment matrix $X\in R^{5\times 40}$ encodes five features across 40 consecutive frames, enabling temporal modeling of pose dynamics with minimal computational overhead.

### 3.3. Classification Model: Soft-Voting Ensemble

As illustrated in Figure 4, the proposed long-lie identification system adopts a modular architecture designed for real-time execution on edge devices. The process begins with thermal video capture, followed by 2D pose estimation using MediaPipe Pose. From the extracted keypoints, handcrafted postural features are generated to capture geometric cues relevant to fall and immobility detection.

To maximize robustness and generalization, a soft-voting ensemble classifier is employed. Rather than relying on a single model, the system evaluates a pool of candidate classifiers based on cross-validated precision, recall, and F1 score. The top three performing classifiers are automatically selected to form the ensemble. This model-agnostic selection ensures adaptability to future updates or datasets without being restricted to pre-defined algorithms.

Each selected classifier produces a class probability distribution over input samples. These probabilities are averaged to obtain an ensemble confidence score:(18)$$P_{\mathrm{ens}}\left(c_{i}|x\right)=\frac{1}{N}\sum_{k=1}^{N}P_{k}\left(c_{i}|x\right)$$
where $P_{k}\left(c_{i}|x\right)$ is the probability from the *k*-th classifier for class $c_{i}$, and N=3 is the total number of selected models. The final prediction is made by(19)$$\hat{y}=\arg\max\left(P_{\mathrm{ens}}\left(c_{i}|x\right)\right)$$

This ensemble mechanism enables the system to integrate diverse decision boundaries while maintaining computational efficiency, making it suitable for deployment on embedded systems, such as the Raspberry Pi 5. The decision to adopt a soft-voting ensemble was based on a comparative evaluation of five classifiers: K-Nearest Neighbors (KNN), Support Vector Classifier (SVC), Multi-Layer Perceptron (MLP), Decision Tree (DT), and Logistic Regression (LR). These models were selected for their efficiency and compatibility with edge hardware. As shown in Table 3, the ensemble combining KNN, SVC, and MLP outperformed all individual models in terms of accuracy, precision, recall, F1 score, and specificity. While more complex architectures such as deep convolutional networks or transformers have been used in related work, [7,8] they require GPU acceleration and are not feasible for lightweight, real-time deployment on resource-constrained devices like Raspberry Pi. In contrast, the soft-voting ensemble balances high classification performance with low inference latency, making it a suitable choice for our privacy-aware, embedded, long-lie monitoring system.

### 3.4. Long-Lie Detection Logic

To determine whether a fall leads to a critical long-lie condition, the proposed system incorporates a post-classification temporal monitoring module. Unlike conventional fall detection systems that issue alerts immediately after a fall is detected, our system continues monitoring the subject’s posture over time to assess prolonged immobility. Once a sample is classified as a “Fall”, the system begins tracking body orientation across consecutive frames. A potential long-lie event is flagged when two conditions are simultaneously met: (1) the subject remains in the fallen state for at least 15 min, and (2) body orientation variation across that duration remains below 15%. This dual-criterion approach ensures that only high-risk, sustained immobility cases are identified, while minor adjustments or recovery attempts are not misclassified. The logic is formalized in Algorithm 1.
**Algorithm 1:** Long-lie Identification1:**if** Status == “Fall” **then**2:    Count Fall_current_time3:    $Body\_ort\_change⇐\left(\left|\frac{body\_ort-body\_ort\_old}{body\_ort\_old}\right|\right)\times 100\%$4:    **if** Fall_current_time ≥ LongLie_Time_Threshold **then**5:        Longlie_Detected ⇐ True6:        **if** Body_ort_change > 15% **then**7:           LongLie_Current_time ⇐08:           Longlie_Detected ⇐ **False**9:           Status ⇐ *“Moving”*10:        **end if**11:    **end if**12:**else**13:    LongLie_Current_time ⇐014:    Fall_current_time ⇐015:    Longlie_Detected ⇐ **False**16:**end if**

The orientation change is calculated as(20)$$Body\_ort\_change=\left(\left|\frac{body\_ort-body\_ort\_old}{body\_ort\_old}\right|\right)\times 100\%$$

In this formulation, *body_ort* represents the ACB angle, defined as the inclination angle formed by a lower body COG, the hip center, and the upper body COG. This angle is computed using Equation (Equation 17) and serves as a scalar descriptor of sagittal body posture. Since it is a single angular feature rather than a multi-dimensional vector, the relative change in Equation (Equation 20) does not require normalization or feature weighting. A fixed threshold of 15% is applied directly to the scalar change value to determine whether the posture remains sufficiently stable to qualify as a long-lie condition. The use of a 15-min monitoring threshold is intentionally conservative, reflecting the nature of our experimental setup with healthy participants in a controlled environment. While clinical studies suggest that adverse consequences can begin within minutes after a fall [2,5,11,12,13], the longer threshold was chosen to reduce false alarms and validate system behavior under realistic, privacy-preserving deployment scenarios. This rule-based logic, which combines time-based monitoring and posture stability, is computationally lightweight and well-suited for implementation on embedded platforms. It enhances long-lie detection specificity by filtering out transient inactivity and allowing timely alerts only for prolonged, motionless conditions.

## 4. Experimental Results

This section presents the real-time implementation, evaluation, and validation of the proposed long-lie detection system using thermal imaging on a resource-constrained edge device. The assessment focuses on classification performance, latency, and system robustness under various environmental and lighting conditions with healthy volunteers.

### 4.1. Real-Time System Deployment on Edge Device

To assess its real-world applicability, the complete pipeline was deployed on a Raspberry Pi 5 (Raspberry Pi Foundation, Cambridge, UK), chosen for its balance between computational capability and energy efficiency. The device features a 64-bit ARM Cortex-A76 CPU and runs Debian 12 (64-bit). A TOPDON TC001 thermal camera (Topdon Technology Co., Ltd., Shenzhen, China) with a resolution of 256×192 pixels and a lightweight design (30 g) was integrated into the system to ensure privacy-preserving monitoring. Both the Raspberry Pi and the camera were mounted on a portable tripod stand with adjustable height (up to 170 cm) and powered by a 5 V/3 A rechargeable battery, enabling flexible and unobtrusive deployment in indoor environments such as homes and clinics. The prototype setup is illustrated in Figure 5.

### 4.2. Classifier Evaluation and Inference Performance

Table 3 presents a detailed comparison of performance metrics across five individual classifiers—KNN, SVC, MLP, DT, and LR—alongside the proposed soft-voting ensemble method. The evaluation metrics include accuracy, precision, recall, specificity, F1 score, the area under the ROC curve (AUC), and the average time of execution per prediction (ATE). The proposed ensemble classifier, which integrates the three best-performing models (KNN, SVC, and MLP), consistently outperforms the individual classifiers across all primary evaluation metrics—achieving 0.98 in accuracy, precision, recall, and F1 score, with a specificity of 0.99 and an AUC of 0.993. These results not only highlight the predictive capabilities of each constituent model but also demonstrate the synergistic effect of ensemble voting in improving overall classification robustness compared with standalone models.

To further contextualize the contribution of our approach, Table 4 provides a comparative overview of representative vision-based fall detection methods in the literature. While many existing methods achieve high accuracy in detecting fall events, few support real-time deployment on edge hardware, and even fewer explicitly address the long-lie scenario.

Our system uniquely combines high accuracy with real-time edge deployability and explicit long-lie tracking. It utilizes thermal input to ensure privacy, incorporates lightweight, handcrafted features for efficiency and interpretability, and employs a soft-voting ensemble for robust fall classification. In contrast to deep learning methods, which typically require high computational resources and offer limited transparency, our approach is explainable and optimized for practical deployment on low-power platforms, such as the Raspberry Pi.

### 4.3. Model Behavior Visualization

Figure 6 illustrates the confusion matrices of five individual classifiers—KNN, SVC, MLP, DT, and LR—along with the proposed soft-voting ensemble model. These visualizations provide detailed insight into the classification behavior and error patterns of each approach.

From the confusion matrices, it is evident that each individual model exhibited distinct performance characteristics. KNN (Figure 6a) demonstrated well-balanced predictions with relatively low misclassification, particularly for fall-related segments. SVC (Figure 6b) and MLP (Figure 6c) also achieved high overall accuracy but were slightly less stable in classifying ambiguous cases near class boundaries. Decision Tree (Figure 6d) showed signs of overfitting, as reflected in its high sensitivity but comparatively lower specificity—suggesting it may have learned patterns that do not generalize well across all samples. Logistic regression (Figure 6e), while efficient, tended to underperform on more complex samples, especially those representing subtle postural transitions.

In contrast, the soft-voting ensemble classifier (Figure 6f) exhibited the most balanced classification performance across all classes, effectively minimizing both false positives and false negatives. This improvement is attributed to the ensemble’s ability to integrate the complementary strengths of its constituent models. For example, KNN’s sensitivity to neighborhood patterns, SVC’s margin-based separation, and MLP’s nonlinear feature learning are harmonized through probabilistic averaging, resulting in a more robust and generalizable decision boundary. These results further emphasize the strength of the ensemble strategy—not merely in achieving higher accuracy but in producing more consistent and reliable predictions across diverse scenarios. This is particularly critical for long-lie detection, where misclassifying a true immobility event (false negative) could delay emergency response, while false positives may cause unnecessary alarms. Overall, the confusion matrix analysis provides compelling evidence that the ensemble classifier delivers a practical balance between performance and reliability, making it a strong candidate for real-time, safety-critical applications such as fall aftermath monitoring in elderly care.

### 4.4. Real-Time Validation with Healthy Subjects

To evaluate the real-time applicability of the proposed system, experimental trials were conducted with two healthy adult volunteers (mean age: 35.5±1.5 years; mean height: 158±8 cm; mean weight: 55 kg). Each subject performed a variety of ADLs—such as sitting, lying, standing, crawling, and walking—along with simulated fall scenarios. The simulated falls were categorized into forward, backward, sideways, and collapsing falls. In total, the dataset used for testing in this phase comprised over 40 min of thermal video, segmented into labeled clips of ADLs and fall responses.

Recordings were conducted in two distinct indoor environments: (1) a dimly lit living room and (2) a brightly illuminated laboratory. In the living room setting (Figure 7), the system successfully distinguished between recoverable falls and sustained immobility. For instance, when the subject remained in a fixed posture with minimal movement, a long-lie condition was flagged after a 15-min observation period with body orientation variation below 15%. While this threshold may be longer than clinical recommendations, it was conservatively chosen to minimize false positives during simulation with healthy subjects, in line with previous long-lie literature [2,5,11,12,13].

In the laboratory setting (Figure 8), despite occasional occlusions or non-frontal orientations, the system retained its ability to track movement trends. Notably, the posture monitoring logic was still able to correctly identify recovery motions—such as repositioning or crawling—as indicators of non-long-lie events. However, we acknowledge that such behaviors (e.g., crawling without successful recovery) could still reflect situations requiring assistance. Addressing this limitation requires deeper behavior modeling beyond orientation-based immobility alone, which we identify as an important area for future development.

## 5. Discussion

The primary objective of this study was to develop a lightweight, privacy-preserving system capable of detecting long-lie conditions following falls using thermal imaging and a soft-voting ensemble classifier. The proposed approach demonstrated high accuracy, precision, recall, and specificity in both offline evaluation and real-time deployment. By combining three complementary classifiers—KNN, SVC, and MLP—into a model-agnostic ensemble, the system minimized classification errors and achieved robustness across various postural conditions. Analysis of confusion matrices (Figure 6) revealed that each individual classifier had distinct strengths and limitations. KNN delivered balanced predictions with low misclassification rates; SVC and MLP provided competitive but slightly less consistent results; DT suffered from overfitting; and LR showed moderate generalization capabilities. The ensemble classifier outperformed all individual models by integrating their complementary strengths through soft voting to produce more reliable and generalized decisions. These findings confirm the benefit of classifier fusion, especially when working with interpretable, handcrafted features under constrained computational resources.

While advanced deep learning architectures such as CNNs, transformers, and autoencoders have demonstrated strong performance in vision-based fall detection tasks, they often require substantial computational resources, large labeled datasets, and GPU-accelerated environments. These requirements conflict with the goals of privacy preservation, edge deployability, and interpretability. In contrast, our use of a soft-voting ensemble based on lightweight classifiers (KNN, SVC, MLP) offers a practical tradeoff between performance and efficiency. It enables real-time inference on resource-constrained devices, such as the Raspberry Pi, without compromising detection accuracy. Moreover, the use of handcrafted geometric features allows domain-level interpretability, which is particularly valuable for healthcare applications where model decisions need to be explainable to non-technical users.

In the real-time evaluation, the system was able to effectively differentiate between typical ADLs, transient falls, and critical long-lie incidents. Trials conducted in both dimly lit and brightly illuminated rooms confirmed that the thermal-based system maintained consistent detection performance independent of ambient light—reinforcing the practicality of thermal sensing for unobtrusive in-home monitoring. Moreover, the deployment on a Raspberry Pi 5 validated the computational efficiency of the entire pipeline, confirming the system’s viability for embedded, cloud-free implementation. The use of a 15-min threshold for long-lie identification was a conservative choice based on two considerations: (1) the ethical constraints of involving healthy participants in prolonged immobility and (2) the goal of minimizing false positives during early-stage testing. While shorter clinical thresholds (e.g., 1–5 min) may be more appropriate in emergency scenarios, our architecture supports easy reconfiguration for real-world deployment with elderly users. Similarly, the 15% posture variation threshold was derived empirically to balance sensitivity and specificity in distinguishing between static immobility and voluntary recovery attempts.

To the best of our knowledge, this is the first thermal-based long-lie detection system that integrates handcrafted geometric features, pose estimation from a reduced keypoint set, and model-agnostic ensemble classification tailored for edge devices. While prior research has focused on fall detection or utilized deep neural networks that are unsuitable for embedded platforms, our approach offers a unique balance between interpretability, efficiency, and deployment readiness. Furthermore, we introduce a rule-based immobility monitoring logic to track postural variation over time, enabling post-fall risk assessment without the need for additional sensors. This contribution directly addresses the research gap, particularly regarding the need for practical, long-life detection systems that preserve privacy, operate in real-time, and are adaptable to non-invasive sensing environments. Our system is also among the few that combines both pre-fall and post-fall cues—using pose and motion data to infer long-lie states.

Despite its strengths, the system has several limitations. First, skeleton keypoint extraction from thermal images is sensitive to occlusion and side-facing orientations. The current method mitigates short-term landmark disappearance using confidence-based filtering, backward interpolation (Equations (Equation 3) and (Equation 4)), and temporal smoothing, which are effective against minor occlusions such as motion blur or partial self-occlusion. However, severe occlusions—caused by furniture, extreme body curling, or low camera angles—can lead to incomplete or noisy pose estimation. In such cases, the system conservatively interprets unreliable input as a movement to avoid false alarms, which may result in missed long-lie detections. Similarly, crawling is currently interpreted as active movement and excluded from the long-lie classification, potentially overlooking individuals who remain mobile despite being incapacitated. Despite the risk of keypoint loss during prolonged static postures, we observed that the thermal appearance of the body in a lying position remains stable across frames. This thermal consistency—combined with the fixed camera perspective and constant environmental conditions—facilitates persistent keypoint visibility over time. In our dataset, long-lie sequences did not result in the systematic disappearance of keypoints; instead, pose estimators maintained a stable set of upper body and hip landmarks throughout. Furthermore, the temporal filtering mechanism compensates for occasional low-confidence detections, ensuring that isolated frame-level dropout does not lead to fragmentation of the pose sequence. Future work will address these limitations by incorporating motion pattern classifiers and dynamic pose models to improve robustness under challenging visual conditions.

Second, sensor placement plays a critical role in coverage and reliability. Our current tripod-mounted, side-angled setup was selected for its portability and suitability for indoor testing; however, a ceiling-mounted, top-down configuration may improve the visibility of all limbs and reduce occlusion—especially during rotational motion or transitions. We plan to evaluate such setups in future iterations. Third, the dataset used in this study consisted of simulated falls performed by healthy adults, which may not fully reflect the behavioral and biomechanical characteristics of older adults. The body orientation, reaction times, and recovery attempts of elderly individuals may differ significantly. Therefore, future validation with elderly participants in real-world environments is essential to ensure clinical relevance. Finally, while our system focuses on skeletal-based features, integrating physiological monitoring (e.g., heart rate or respiration) could enhance detection confidence and allow earlier intervention. Thermal imaging opens possibilities for passive vital sign estimation, which we intend to explore in subsequent phases of this research.

Overall, the proposed system demonstrates the feasibility of combining privacy-preserving sensing, interpretable features, and efficient ensemble classification into a deployable, edge-compatible solution. It addresses key concerns in fall aftermath monitoring, including ethical sensing, latency, and real-time responsiveness. By bridging the gap between experimental robustness and practical deployment, this work contributes to the growing field of unobtrusive elderly care technologies. In summary, the system represents a novel and practical approach to long-lie detection that is both privacy-aware and resource-efficient. By addressing key limitations through future enhancements—such as advanced behavior recognition, top-down sensing, and elderly-focused validation—this system has strong potential to be translated into real-world applications and integrated into homecare or clinical monitoring platforms.

## 6. Conclusions

Timely detection of long-lie incidents is essential to reducing the risk of severe health complications among elderly individuals living independently. This study proposed and validated a privacy-preserving long-lie detection system based on thermal imaging and a soft-voting ensemble classifier (KNN, MLP, SVC) deployed on a resource-constrained single-board computer. The system achieved strong classification performance, with accuracy, precision, recall, and F1 score of 0.98, specificity of 0.99, and AUC of 0.993—demonstrating its capability to distinguish long-lie conditions from ordinary falls and activities of daily living in real time. Despite its promising results, the study acknowledges several limitations. The system’s detection performance may be affected by low-resolution thermal imaging, partial occlusions, and atypical movement patterns such as crawling. Moreover, the evaluation was conducted solely on healthy participants performing simulated scenarios, which may not fully capture the variability in posture and recovery behaviors observed in elderly populations. The current logic also classifies crawling as movement, potentially overlooking cases where assistance is still required. Future work will focus on addressing these limitations by incorporating higher-resolution thermal sensors, refining behavior recognition algorithms to detect complex recovery attempts, and validating the system with elderly users in real-world settings. Additionally, integrating passive vital sign monitoring could further enhance the system’s ability to assess user status and support timely medical intervention. In conclusion, this work presents a practical, interpretable, and deployable solution for long-lie detection that balances privacy preservation, computational efficiency, and detection accuracy. With further development and validation, the proposed system holds strong potential to be integrated into intelligent in-home monitoring platforms for elderly care.

## Figures and Tables

**Figure 1 sensors-25-03836-f001:**
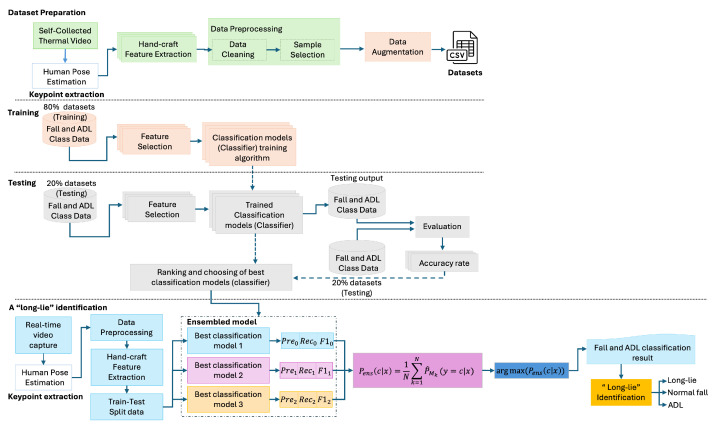
The block diagram of the real-time long-lie detection system. The process for long-lie identification starts with dataset preparation, including keypoint extraction, feature generation, and data preprocessing to create a labeled dataset. Classification models are trained and tested using fall and activities of daily living (ADL) living data to evaluate their performance. The final stage uses an ensemble approach to identify long-lie incidents based on the best-performing models.

**Figure 2 sensors-25-03836-f002:**
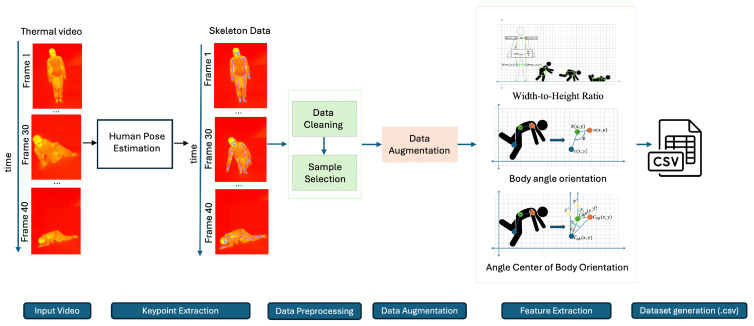
The workflow for dataset generation involves processing thermal video frames using human pose estimation algorithm to extract skeleton data, followed by data cleaning, sample selection, temporal data augmentation, and feature extraction. The final features are compiled into a structured CSV dataset used for model training and testing.

**Figure 3 sensors-25-03836-f003:**
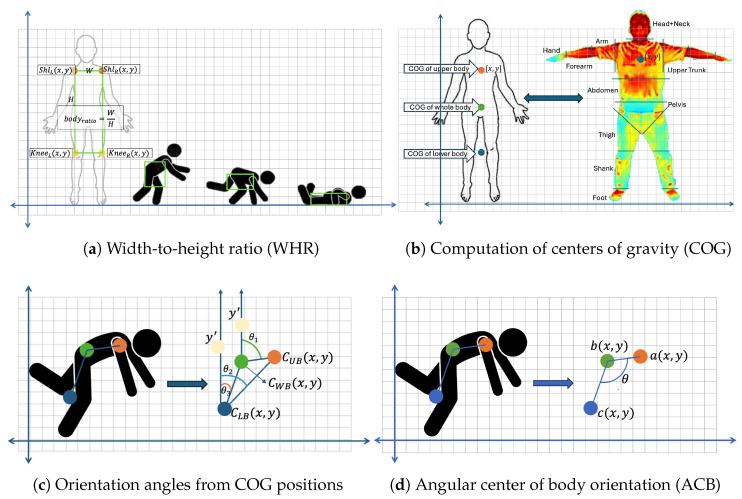
Illustration of handcrafted feature extraction: (**a**) Projected width-to-height ratio (WHR) calculated from shoulder-to-shoulder width and shoulder-to-knee height, used as a posture descriptor in 2D thermal images, (**b**) computation of multiple centers of gravity (COG), (**c**) orientation angles derived from COG positions, and (**d**) calculation of angular center of body orientation (ACB).

**Figure 4 sensors-25-03836-f004:**
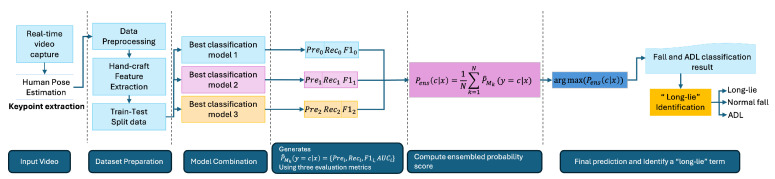
The ensemble architecture for long-lie identification system.

**Figure 5 sensors-25-03836-f005:**
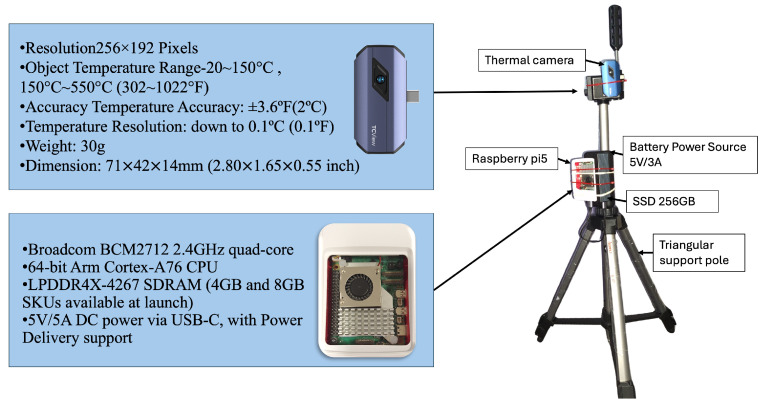
Prototype of the long-lie identification system, featuring a thermal camera and Raspberry Pi 5 mounted on a tripod with a 5 V/3 A battery pack. The system operates in real time while preserving user privacy.

**Figure 6 sensors-25-03836-f006:**
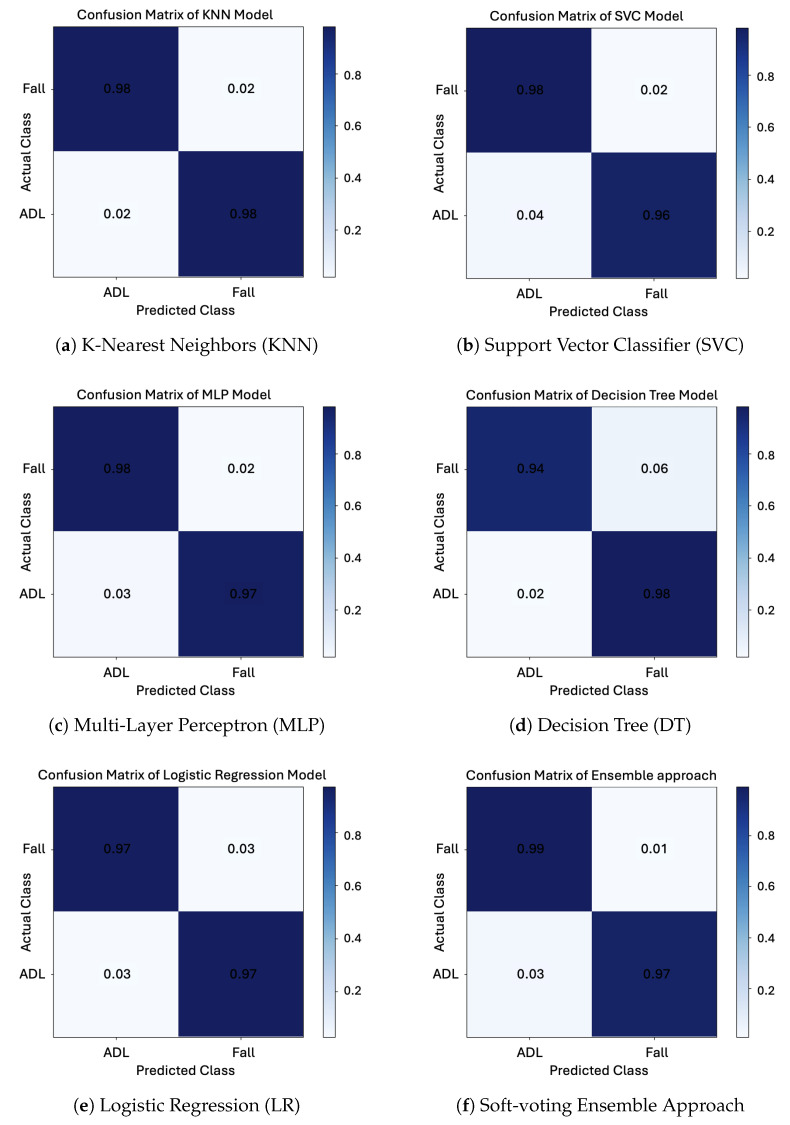
Confusion matrices of various classification models for performance evaluation: (**a**) KNN, (**b**) SVC, (**c**) MLP, (**d**) DT, (**e**) LR, and (**f**) Soft-voting Ensemble. These matrices reflect actual versus predicted class distributions.

**Figure 7 sensors-25-03836-f007:**
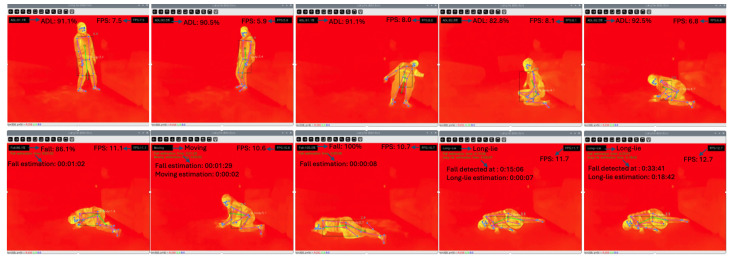
Long-lie detection in real-time from thermal input. Subject remains immobile after a fall beyond the 15-min threshold.

**Figure 8 sensors-25-03836-f008:**
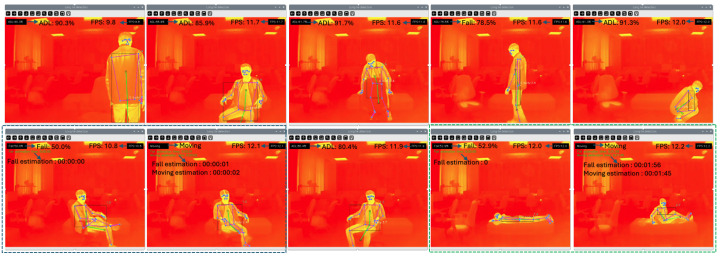
Recovery detection after fall. The subject repositions after impact, preventing false long-lie detection.

**Table 1 sensors-25-03836-t001:** Participant details for dataset collection including age, height, weight, and gender.

No.	Age (Years)	Height (cm)	Weight (kg)	Gender (F/M)
1	34	150	55	F
2	37	168	75	M
3	36	160	64	M
4	30	160	55	M
5	28	168	70	F
6	26	158	49	F
7	25	163	60	F
8	24	185	70	M
9	29	158	68	F
10	25	174	68	M

**Table 2 sensors-25-03836-t002:** Aggregated detection rate of selected keypoints from two thermal infrared videos. A keypoint is considered valid if its confidence exceeds 0.5.

Keypoints	valid frames	total frames	Detection Rate (%)
Nose	3710	3742	99.15
Left Shoulder	3710	3742	99.15
Right Shoulder	3710	3742	99.15
Left Hip	3710	3742	99.15
Right Hip	3710	3742	99.15
Left Knee	3010	3742	80.44
Right Knee	3050	3742	81.51
Average ± SD	–	–	93.96 ± 8.22

**Table 3 sensors-25-03836-t003:** Comparison of performance metrics across individual classifiers and the proposed ensemble method. The proposed ensemble, combining KNN, SVC, and MLP, achieves the highest accuracy and robustness with a modest execution time.

No.	Models	Acc ^1^	Prec ^2^	Recall	Specificity	F1 Score	AUC ^3^	ATE (s) ^4^
1	KNN	0.97	0.98	0.98	0.98	0.98	0.993	0.1460
2	SVC	0.97	0.97	0.97	0.98	0.97	0.982	0.0331
3	MLP	0.97	0.97	0.97	0.98	0.97	0.989	0.0045
4	DT	0.95	0.96	0.96	0.94	0.96	0.959	0.0012
5	LR	0.97	0.97	0.97	0.97	0.97	0.984	0.0055
6	Ours	0.98	0.98	0.98	0.99	0.98	0.993	0.2687

^1^ Acc: Accuracy. ^2^ Prec: Precision. ^3^ AUC: Area Under the ROC Curve. ^4^ ATE: Average Time of Execution per prediction.

**Table 4 sensors-25-03836-t004:** Comparative analysis of vision-based fall detection methods using thermal, RGB cameras, and mmWave radar. Privacy: high = de-identified human forms, low = identifiable images. Real-time: suitable for online/faster-than-video-rate processing. Edge Deployable: feasible on edge hardware (e.g., SBCs). Long-Lie Estimation: explicit immobility tracking.

Method	Sensor Type	Algorithm	Acc ^1^	Pri ^2^	RT ^3^	ED ^4^	LLE ^5^	Notes
Ours	Thermal cam (256×192)	Ensemble (KNN, SVC, MLP)	98%	High	Yes	Yes	Yes	Real-time on edge; low computation; supports long-lie tracking
Lau et al. [43]	RGB	Attention-based GRU	96.2%	Low	Yes	No	No	Deep model; high computation; no hardware info; likely tested offline
Elshwemy et al. [22]	Thermal	SRAE	-	High	No	No	No	Offline only; moderate computation; platform not specified
Rezaei et al. [21]	Low-res Thermal	CNN + manual features	97.9%	High	Yes	Yes	No	Edge-intended; low computation; no device reported
Zhang et al. [6]	mmWave Radar	Rule-based (point cloud)	-	High	Yes	Yes	Yes	Rule-based logic; medium computation; runs on desktop CPU only
Zhong et al. [20]	Thermal vision	CNN + RBFNN	98.39%	High	Yes	No	No	Likely tested offline; moderate-to-high computation; no hardware info

^1^ Acc: Accuracy. ^2^ Pri: Privacy protection. ^3^ RT: Real-time implementation. ^4^ ED: Edge deployable. ^5^ LLE: Long-lie estimation.

## Data Availability

The datasets generated and analyzed during the current study are available from the corresponding author on reasonable request.

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
