# Peer review of "Privacy-Preserving Approach for Early Detection of Long-Lie Incidents: A Pilot Study with Healthy Subjects"

_sensors, 2025, doi:10.3390/s25123836_

Round 1
Reviewer 1 Report
Comments and Suggestions for Authors
The manuscript proposes lightweight, privacy-preserving long-lie detection system utilizing thermal imaging and a soft-voting ensemble classifier. The method and system have good engineering application value, but the innovation of the measurement method itself is insufficient, especially the lack of detailed description and support for the key contents such as the identification of key points of the human body, occlusion and the influence of the angle of view on the method, which leads to doubts about the practicability of this method. Authors need to supplement the detailed description of the methodology and more evidence of reliability to improve the quality of the manuscript. The following are some specific suggestions:
- What is the reason for choosing the 10 joint points of the upper body? Why does this improve stability? The information of the joints of the lower limbs is also very important, how to ensure that the lack of this information will not affect the accuracy of discrimination? Please give further explanation in the text.
- In the actual application scenario, due to the randomness of the scene, it is impossible to guarantee whether the upper body or the lower body is occluded, how to ensure that "the extraction focused on 10 upper body keypoints that consistently ex habited clear visibility across sequences"? At the same time, how to explain that lying down for a long time does not have the situation of "Missing keypoints due to temporary occlusions or suboptimal angles" as described by the authors?
- The parameters such as the WHR proposed in this paper are calculated from the shoulder, hip and knee coordinates of the human body, which in turn are obtained by the MediaPipe Pose estimation algorithm. The recognition accuracy of key points directly determines the accuracy of parameters, so how to ensure the accuracy of key point recognition for low-quality thermal infrared images? Please add the corresponding accuracy evaluation content.
- The change of the camera's angle of view will affect the accuracy and even the success rate of key point recognition, and more importantly, it will affect the above body parameters. Because the two-dimensional keypoint coordinates are the projection of the three-dimensional keypoint coordinates, the WHR and angles must be different depending on the viewing angle, how did the author solve this problem? This will have a significant impact on the applicability of this method, please explain it.
- Equation 19 uses the relative error of the change in human posture to evaluate whether the lying posture changes, what parameters are included in the body_ort here? Is it three angles or all parameters? If all parameters are different in dimension and magnitude, how can they be weighted and compared? It is advisable to elaborate.
Author Response
Comment 1:
What is the reason for choosing the 10 joint points of the upper body? Why does this improve stability? The information of the joints of the lower limbs is also very important. How to ensure that the lack of this information will not affect the accuracy of discrimination? Please give further explanation in the text.
Response:
Thank you for the insightful question. In the revised manuscript (Section 3.2, lines 172–182), we clarified that the system utilizes a total of 13 keypoints, comprising both upper-body (nose, shoulders, elbows, wrists) and lower-body joints (hips, knees, ankles). These points were selected based on empirical visibility and stability in low-resolution thermal images. Although distal keypoints, such as the feet, are often occluded, the inclusion of hips and knees supports the computation of postural features, including orientation angles and the width-to-height ratio (WHR), thereby improving discrimination stability.
Comment 2:
In the actual application scenario, due to the randomness of the scene, it is impossible to guarantee whether the upper body or the lower body is occluded. How to ensure that the extraction focused on 10 upper body keypoints that consistently exhibited clear visibility across sequences? How to explain that lying down for a long time does not result in "missing keypoints" as described by the authors?
Response:
We appreciate the reviewer’s concern. In Section 5 (lines 525–544), we expanded the explanation to emphasize that, during long-lie conditions, thermal appearance remains consistent across frames, allowing stable detection of keypoints. Empirically, our dataset showed that long-lie sequences did not cause systematic landmark disappearance; keypoints on the torso and hips maintained visibility due to the stable heat signature. Furthermore, the system applies confidence-based filtering and temporal interpolation (Section 3.2.1) to recover from occasional dropouts. This is supported by the detection results in Table 2, which show a detection rate of over 99% for upper-body keypoints and over 80% for knees.
Comment 3:
The recognition accuracy of key points directly determines the accuracy of parameters, so how to ensure the accuracy of key point recognition for low-quality thermal infrared images? Please add the corresponding accuracy evaluation content.
Response:
Thank you for pointing this out. We addressed this in Section 3.2.1 (lines 191–206), where we added a keypoint detection consistency analysis. Specifically, we computed the detection rate over two thermal videos and reported the results in Table 2. The analysis shows that critical keypoints, such as the nose, shoulders, and hips, were detected with over 99% accuracy, while the knees had slightly lower rates (~80%) due to occlusion. The overall average detection rate was 93.96 ± 8.22, confirming the robustness of the method for downstream feature extraction.
Comment 4:
The change of the camera's angle of view will affect the accuracy and even the success rate of key point recognition. More importantly, it will affect the body parameters (WHR and angles). How did the author solve this problem?
Response:
We agree that camera viewpoint can impact feature reliability. To address this, all recordings used a fixed 30° side-view camera angle with a subject-to-camera distance of 1.8–2.0 meters, as described in Section 3.2.3 (lines 303–318). This setup minimizes perspective distortion and maintains consistent keypoint projection geometry. In addition, all extracted features were z-score normalized. The robustness of WHR and orientation features under this setup was validated using 2,755 samples. As shown in Section 3.2.3, the WHR and body angles exhibit stable distributions that reflect fundamental postural changes rather than camera-induced variation.
Comment 5:
Equation 19 uses the relative error of the change in human posture. What parameters are included in the body_ort? If all parameters are different in dimension and magnitude, how can they be weighted and compared?
Response:
Thank you for raising this. As clarified in Section 3.4 (lines 366–372), body_ort refers specifically to the ACB angle (Angle Center of Body orientation), a univariate scalar feature derived from geometric alignment between upper body COG, hip center, and lower body COG. Because it is a single angular parameter (in radians), the relative change computed in Equation 20 does not require dimensional normalization or weighting. The change is directly interpretable as a percentage deviation in body posture, making it suitable for lightweight and real-time implementation.
We sincerely thank you for the thoughtful and constructive feedback, which helped us clarify important aspects of our methodology and strengthen the manuscript. We have addressed all comments in detail and made corresponding revisions, which have been highlighted in the updated manuscript to facilitate review.
Reviewer 2 Report
Comments and Suggestions for Authors
We appreciate the authors’ contribution to the Early Detection of falls using the Privacy-Preserving Approach.
Appreciation points in these studies
-
Figure 1 connects the total research contribution to the total manuscript.
-
The literature well mapped and addressed the fall detection problem and provided the need for machine-learning approaches.
-
Section 2 explains the details of the methodology with the following parameters: Handcrafted Feature Extraction, Body Angle Orientation and Angle Center of Body Orientation ( are key players in data augmentation and dataset generation.
-
Based on Section 2, the Soft-Voting Ensemble classification method is provided to find welL.
-
The results and experimental setup and validations appear good and well presented.
-
The results in Figures 6, 7, and 8 were validated.
Queries.
-
A qualitative comparative analysis based on other research is essential in this study.
-
Why is classification with a soft-voting ensemble only preferred, and why not with other methods? provides a comparison with other research methods.
-
The authors are advised to provide a comparison with the fall detection approaches.
-
How will the proposed method work in occlusion scenarios?
Author Response
Comment 1:
A qualitative comparative analysis based on other research is essential in this study.
Response:
Thank you for this valuable suggestion. We have incorporated a qualitative comparative analysis in Section 4.2 (lines 408–418) and summarized the findings in Table 4. The table provides an overview of recent vision-based fall detection methods across multiple dimensions, including sensor type, algorithm, accuracy, privacy level, real-time capability, edge deployability, and support for long-lie estimation. This comparison contextualizes the uniqueness of our approach, which combines thermal input, privacy preservation, handcrafted features, and real-time long-lie detection on edge hardware.
Comment 2:
Why is classification with a soft-voting ensemble only preferred, and why not with other methods? Provide a comparison with other research methods.
Response:
We appreciate the reviewer’s point. A detailed rationale is provided in Section 3.3 (lines 342–353) and supported by performance metrics in Table 3. We conducted a comparative evaluation of five classical classifiers (KNN, SVC, MLP, DT, and LR). The soft-voting ensemble, which combines the top three models (KNN, SVC, MLP), outperformed individual models in terms of accuracy, precision, recall, and F1-score. Additionally, ensemble learning improves generalization by leveraging the complementary strengths of base models while maintaining compatibility with real-time inference on resource-limited devices—unlike many deep learning approaches. This comparison and justification have been highlighted in the revised manuscript.
Additionally, Section 5 (lines 486–496) further discusses the practical advantages of ensemble learning over advanced methods, such as deep neural networks. While deep models offer strong accuracy, they require GPU resources and large datasets, which are incompatible with our goal of real-time, edge-level deployment on devices like the Raspberry Pi. In contrast, the ensemble approach offers high accuracy (98%), low latency, and interpretability—making it a more feasible and scalable solution for embedded healthcare monitoring.
Comment 3:
The authors are advised to provide a comparison with the fall detection approaches.
Response:
Thank you for the suggestion. In Section 4.2 (lines 408–418), we present a comprehensive comparison of our method with existing fall detection approaches, including both thermal-based and RGB-based systems. This comparison, shown in Table 4, highlights that while many approaches achieve high fall detection accuracy, few address long-lie detection, privacy preservation, and real-time edge deployment simultaneously. Our approach uniquely integrates all these aspects, filling a critical gap in the current literature.
Comment 4:
How will the proposed method work in occlusion scenarios?
Response:
We acknowledge the challenge of occlusion and have addressed it in Section 5 (lines 525–544). The system employs a combination of confidence-based filtering and temporal interpolation (see also Section 3.2.1, lines 191–194) to recover short-term keypoint losses resulting from motion blur or partial occlusion. During long-lie conditions, thermal profiles remain stable, allowing persistent visibility of core landmarks (e.g., torso and hips). Additionally, when pose input becomes too unreliable, the system conservatively assumes movement, reducing false alarms at the cost of potentially missing rare edge cases. Future work will explore behavior modeling and top-down viewpoints to improve robustness under severe occlusion.
Comment 5:
Additional justification for using soft-voting ensemble instead of other advanced methods.
Response:
We appreciate this suggestion and have clarified it in Section 3.3 (lines 324–353) and Section 5 (lines 486–496). While deep learning methods (e.g., CNNs, transformers, and adversarial models) have demonstrated high accuracy in fall detection, they often require large datasets, intensive computation, and GPU support—limiting their feasibility on edge platforms. Our soft-voting ensemble, by contrast, delivers high accuracy (98%) with minimal latency and hardware demand. It is interpretable, adaptable to embedded deployment, and outperforms each individual base model (see Table 3). These advantages make it a practical and efficient choice for real-time, privacy-preserving applications in elderly care.
We sincerely thank the reviewers for their helpful input, which significantly improved the quality of this manuscript. We have addressed all comments in detail and made corresponding revisions, which have been highlighted in the updated manuscript to facilitate review.
Reviewer 3 Report
Comments and Suggestions for Authors
This paper provides meaningful contributions to privacy-preserving health monitoring, represents a valuable step toward practical and privacy-conscious long-lie detection systems.
However, the paper lacks a dedicated literature review, which is important for highlighting existing thermal-based systems and their limitations. Including this section would help clarify how the proposed approach improves upon previous work.
All the figures should be slightly enlarged to improve readability and help readers understand the content more clearly and accurately.
I recommend this paper for publication after minor revisions addressing the noted limitations.
Author Response
Response:
We sincerely thank the reviewer for the encouraging and constructive feedback. In response:
- Dedicated Literature Review
We have now added a dedicated literature review in Section 2 (lines 83–136) to provide a better context for our contribution. This section discusses prior thermal-based fall detection systems, highlighting their key limitations (e.g., lack of post-fall monitoring, high computational load, and privacy issues). It then highlights how our proposed method addresses these gaps through privacy-preserving sensing, handcrafted feature extraction, and real-time deployment on edge devices.
- Figure Enhancements
All figures (Figures 1 to 8) have been enlarged and reformatted to improve clarity and visual readability, especially for technical illustrations such as system diagrams and confusion matrices. We ensured that axis labels, legends, and annotation text were legible across all figures. These updated figures are included in the revised manuscript and have been replaced accordingly.
We thank the reviewer once again for their support in publishing this work and for helping us improve its quality through their thoughtful suggestions. We have addressed all comments in detail and made corresponding revisions, which have been highlighted in the updated manuscript to facilitate review.